# Joint Soft–Hard Attention for Self-Supervised Monocular Depth Estimation

**DOI:** 10.3390/s21216956

**Published:** 2021-10-20

**Authors:** Chao Fan, Zhenyu Yin, Fulong Xu, Anying Chai, Feiqing Zhang

**Affiliations:** 1University of Chinese Academy of Sciences, Beijing 100049, China; fanchao18@mails.ucas.ac.cn (C.F.); xufulong16@mails.ucas.ac.cn (F.X.); chaianying15@mails.ucas.edu.cn (A.C.); zhangfeiqing17@mails.ucas.ac.cn (F.Z.); 2Shenyang Institute of Computing Technology, Chinese Academy of Sciences, Shenyang 110168, China; 3Liaoning Key Laboratory of Domestic Industrial Control Platform Technology on Basic Hardware & Software, Shenyang 110168, China

**Keywords:** monocular depth estimation, self-supervised learning, attention mechanism

## Abstract

In recent years, self-supervised monocular depth estimation has gained popularity among researchers because it uses only a single camera at a much lower cost than the direct use of laser sensors to acquire depth. Although monocular self-supervised methods can obtain dense depths, the estimation accuracy needs to be further improved for better applications in scenarios such as autonomous driving and robot perception. In this paper, we innovatively combine soft attention and hard attention with two new ideas to improve self-supervised monocular depth estimation: (1) a soft attention module and (2) a hard attention strategy. We integrate the soft attention module in the model architecture to enhance feature extraction in both spatial and channel dimensions, adding only a small number of parameters. Unlike traditional fusion approaches, we use the hard attention strategy to enhance the fusion of generated multi-scale depth predictions. Further experiments demonstrate that our method can achieve the best self-supervised performance both on the standard KITTI benchmark and the Make3D dataset.

## 1. Introduction

High-accuracy depth estimation plays a very important role in 3D computer vision, especially for applications in robot navigation [1], autonomous driving [2], virtual reality [3], etc. There are various methods of obtaining depth information, such as laser sensors, or binocular or multi-cameras. However, these might be not available in some cases due to high costs or high environmental demands. Actually, monocular images or videos are more common in most scenes. This has led to the rapid development of monocular depth estimation as only one single camera is required. Although it is essentially an ill-posed problem to obtain depth from an RGB image, recent studies have proved that convolutional neural networks (CNNs) can estimate depth by learning the hidden depth clues from the RGB images. The methods based on deep learning mainly include supervised learning and self-supervised learning. Even though supervised learning methods [4,5,6] can achieve excellent estimation results, the requirement for high-precision ground truth data sets limits on the application of these methods. Self-supervised learning methods [7,8,9,10] construct new supervised signals by using depth predictions as intermediate variables through geometric constraints of monocular video or stereo images.

Self-supervised depth estimation methods based on monocular video have a wider range of applications than approaches based on stereo images due to the fact that the data are more readily available and more common. However, monocular video-based methods need to estimate not only the depth but also the ego-motion [7] during training, which makes the methods prone to introducing more noise. A few special pixels that belong to the occlusion parts or moving objects in the images will seriously affect the accuracy of unsupervised monocular depth estimation. To address these issues, many efforts have focused on improving accuracy by designing new masks [11,12,13], new loss functions [11,14], or using pretrained semantic segmentation networks [15,16]. Another relatively more direct approach is the improvement of the network’s extraction of depth-dependent features from a new specialized architecture, i.e., strengthening the feature extraction capability of the network [17]. Even with the rapid development of the above methods, the accuracy of self-supervised monocular depth estimation methods is still inferior to that of supervised methods.

In this paper, we propose a new method for exploiting the attention mechanism [18] to improve the precision of self-supervised monocular depth estimation. An example result from our method is shown in Figure 1. The main contributions of our work are as follows:A novel soft attention module that integrates spatial attention and channel attention. The soft attention module not only correlates different non-contiguous regions in the spatial dimension, but also explores the distribution of attention in the channel dimension. In contrast to other current attention methods [19,20], our soft attention module achieves optimal performance with only a small number of parameters.A novel hard attention strategy for fusing the depth estimation results at different scales. Unlike traditional fusion approaches [11,14,21], we assign different attention weights by a hard attention strategy for the generated multi-scale depth predictions, which leads to a better fusion result.Experiments show that our contributions can help us achieve the best self-supervised performance both on the standard KITTI benchmark [2,22] and the Make3D [23] dataset. Either contribution can be easily integrated into many previously well-performing approaches, especially the proposed hard attention strategy.

## 2. Related Works

### 2.1. Monocular Depth Estimation

Many computer vision tasks related to navigation, localization, or 3D scene reconstruction rely heavily on the accuracy of depth estimation. As an alternative to active vision, which suffers from obvious drawbacks such as high cost and limited range, monocular depth estimation, especially based on deep learning, has developed rapidly in recent years. Eigen et al. [4] first used deep learning methods to achieve monocular depth estimation. They implemented depth predictions based on fully supervised learning by integrating a coarse estimation network and a fine estimation network. Liu et al. [24] proposed a method combining a conditional random field (CRF) with deep convolutional networks to strengthen the local smoothness of depth predictions, while further optimizing the edge part of the predictions. Fu et al. [6] reformulated the learning as an ordinary regression problem by the proposed spacing-increasing discretization (SID) strategy. Lee et al. [25] used a patch-wise attention network to incorporate the relationships between neighboring pixels within each local region in the image into the depth estimation. However, these fully supervised deep learning methods can be limited in their applications due to the cost of RGB-D data acquisition.

Self-supervised learning is an effective way to solve the above problems caused by training data. Garg et al. [21] proposed an alternative self-supervised method by using a stereo photometric reprojection warping loss to constrain depth. Godard et al. [14] improved the performance of the stereo-based self-supervised methods by further adding left–right consistency loss as a new constraint. Watson et al. [26] enhanced stereo matching in self-supervised depth prediction by introducing depth hints to attenuate the effects of ambiguity reprojections. Even though self-supervised learning models based on stereo image pairs are capable of performing depth prediction with relatively satisfactory accuracy, its training process still requires stereo images, which means that its use will be limited. Since self-supervised learning methods based on monocular video can be implemented with only a single camera, they are more likely to be favored by researchers. Zhou et al. [7] implemented self-supervised learning based on monocular video by jointly training a depth estimation network and a pose network. The estimated depth and the pose between consecutive frames provided the supervised signal during training after the correlation was established by structure-from-motion (SfM). However, self-supervised learning methods based on monocular video require a strict assumption that the scene is stationary except for the camera. To deal with the noise caused by motion in the scene, Vijayanarasimhan et al. [12] designed multiple motion masks for depth predictions. Another effective way to mask the moving objects is making use of trained semantic models [15,16,27,28]. Other advances in recent years have focused on improving the accuracy of self-supervised training based on monocular video, e.g., discrete depth predictions [20,29,30], handling of occlusion issues [28], improved network architectures [17,31], and using multiple frames at test time [32,33]. Our proposed model based on Monodepth2, named SHdepth, is extended with our proposed ideas.

### 2.2. Attention Mechanism

Inspired by the flexibility of human vision to focus on certain regions of an image, the attention mechanism identifies the input parts that require attention and processes them efficiently [34]. The attention mechanism is divided into hard attention and soft attention according to whether the ‘weight mask’ can be learned or not. Specifically, the weight mask in hard attention is fixed manually. Once the weights are fixed, the network will always focus on the higher weighted parts, while reducing the impact of other parts. In contrast, the weight mask in soft attention is continuously learned and updated during training so that the model can automatically adjust the attention weights according to different datasets or different tasks. Furthermore, soft attention is usually implemented by an additional designed module that can be readily integrated into CNN models. Hu, Shen, and Sun [35] proposed squeeze-and-excitation networks (SENets), which implemented adaptive weighting by global average pooling in the channel dimension. Woo et al. [36] extended the SENet by adding max-pooling when calculating the channel attention. They also designed a new spatial attention block and merged it with channel attention in sequence, namely a convolutional block attention module (CBAM). Wang et al. [37] presented a generalized non-local operation block that can be directly plugged into many computer vision architectures to capture long-range dependencies. The non-local operation has been an early successful attempt of self-attention on visual tasks. Gao et al. [19] devised an attentional separation-and-aggregation network (ASANet) that can distinguish whether the pixels belong to static or dynamic scenes via the attention mechanism. By using the ASANet as an encoder, the camera’s ego-motion and the scene’s dynamic motion field can be estimated. Johnston and Carneiro [20] used self-attention to improve contextual reasoning in their self-supervised monocular trained model. The experiment proved their hypothesis that the context for the prediction of a pixel depth may be in non-contiguous positions. Inspired by the above attention mechanism, we hypothesize that the attention maps generated by self-attention may require additional weight assignment in the channel dimension due to the different importance of non-adjacent regions on different feature maps for the depth estimation of a pixel.

## 3. Method

In this section, we focus on showing the implementation details of our contributions, including soft attention and hard attention. We first introduce the overall framework, and then describe and analyze the soft attention module that can be integrated into a U-Net [38] architecture. Finally, we describe how the hard attention strategy is implemented through a new loss function in conjunction with the principles of self-supervised training for monocular depth estimation.

### 3.1. Overall Framework

An overview of the SHdepth framework is presented in Figure 2. Our SHdepth is based on the Monodepth2 model, which uses a standard U-Net to predict depth. Specifically: (1) the soft attention module we proposed is at the end of the ResNet [39] encoder module. As a result, the soft attention module can further encode the network to extract more appropriate features and has less computational effort with an input resolution of $H_{o}/32\times W_{o}/32$ ($H_{o}$ denotes the height of the original input RGB image and $W_{o}$ denotes the width). (2) The hard attention strategy is adopted after the photometric loss [11] calculation for multi-scale depth predictions and eventually achieves the fusion of multi-scale results through a new loss function.

### 3.2. Soft Attention Module

Inspired by the success of the attention mechanism in deep learning, we propose a soft attention module to capture long-range dependencies in the spatial dimension and explore the distribution of attention in the channel dimension. In general, the soft attention module consists of two parts, as shown in Figure 3. The first part is a spatial attention that is similar to the self-attention form in [18]. The second part is a channel attention integrated with a residual connection. Finally, we arrange the two submodules in a spatial-channel order. Moreover, the number of parameters for our soft attention module is only 0.82 M.

#### 3.2.1. Spatial Attention

We represent the input feature map of spatial attention as tensor X of size B×C×H×W. Since the context for the prediction of a pixel depth may be in non-contiguous positions, we first employ spatial attention to capture the relationship between pixels in the non-adjacent regions. Specifically, we use a modified non-local operation in the embedded Gaussian version inspired by [37]. There are two differences compared to the original non-local operations. The first one is the removal of the residual structure in consideration of better transfer of information in the attention maps. The other one is not to compress the number of channels in the first three independent 1×1×1 convolution layers. It is well known that matrix multiplication in spatial attention has a high computational cost, especially when the input feature map size is too large. In fact, we do not reduce the cost by compressing the number of channels, because we have decreased the size of the input feature map to 1/32 of the original RGB input image by placing the spatial attention at the end of the ResNet. Finally, we can compute the output of spatial attention using the following equation:(1)$$M_{s}=softmax\left(W_{\theta }XX^{T}W_{\phi }^{T}\right)W_{g}X$$
where $W_{\theta }$, $W_{\phi },$ and $W_{g}$ are the weight matrices implemented as 1×1×1 convolutions.

#### 3.2.2. Channel Attention

We feed the output of spatial attention to the channel attention integrated with a residual connection. $M_{s}$ is the input tensor of size B×C×H×W, as shown in Figure 3. We first exploit both average-pooling and max-pooling operations to aggregate the spatial information of the input feature map. Next, we feed the two generated feature maps to a shared multi-layer perceptron (MLP), which includes two fully connected layers and a ReLU activation function. Subsequently, the two separate outputs of the MLP are merged by using element-wise summation. Then, the fused feature map is input into the sigmoid function. Finally, the channel attention maps are generated by the above steps. To summarize, the channel attention is computed as:(2)$$M_{c}=\sigma \left(MLP\left(AvgPool\left(M_{s}\right)\right)+MLP\left(MaxPool\left(M_{s}\right)\right)\right)$$
where σ denotes the sigmoid function and MLP comprises one hidden layer and the ReLU activation function. Note that the parameters in MLP are shared for the two different inputs.

The final output of the soft attention module is then computed as:(3)$$Y=M_{s}+M_{s}⊙M_{c}$$
where ⊙ denotes element-wise multiplication. The size of tensor $M_{c}$ can automatically change from B×C×1×1 to B×C×H×W by the broadcasting mechanism during the element-wise multiplication.

### 3.3. Hard Attention Strategy

#### 3.3.1. Self-Supervised Training

Unlike supervised learning, self-supervised training of depth estimation requires the creation of new supervised signals that are not depth. In fact, we need to use the depth network to predict the depth of the input image without the ground truth depth during training. Similarly to SfMLearner [7], an additional pose network is employed to estimate the pose between adjacent frames. Here we use $p_{t}$ and $p_{t^{\prime }}$ to denote the homogeneous coordinates of pixels in the target frame $I_{t}$ and the source frame $I_{t^{\prime }}$, respectively. Note that the source frame is adjacent to the target frame. Thus, we can formulate the following equation to warp one view into another according to the epipolar geometry.
(4)$$p_{t^{\prime }}\;\sim \;KT_{t\to t^{\prime }}D_{t}\left(p_{t}\right)K^{-1}p_{t}$$
where $D_{t}$ is the depth prediction result in $I_{t}$ and K denotes the camera intrinsics. $T_{t\to t^{\prime }}$ is the relative pose predicted by the pose network. Thus, we can obtain the synthesized target image $I_{t^{\prime }\to t}$ by Equation (4). The per-pixel photometric loss is defined by:(5)$$L_{p}=\underset{t^{\prime }}{\min}\mu \times pe\left(I_{t},I_{t^{\prime }\to t}\right)$$
where pe is the photometric reconstruction error, $t^{\prime }\in \left\{t-1,t+1\right\}$, and μ is a binary mask that filters out pixels that are stationary relative to the camera (see more details below in Equation (8)). We follow [11] to define our photometric error function pe, i.e.,
(6)$$pe\left(I_{a},\;I_{b}\right)=\frac{\alpha }{2}\left(1-SSIM\left(I_{a},\;I_{b}\right)\right)+\left(1-\alpha \right)\|I_{a}-I_{b}{\|}_{1}$$
where α=0.85. Furthermore, in order to optimize the depth prediction around object boundaries, an edge-aware smoothness regularization term [14] is used:(7)$$L_{smooth}=\left|\partial_{x}d_{t}^{*}\right|e^{-\left|\partial_{x}I_{t}\right|}+\left|\partial_{y}d_{t}^{*}\right|e^{-\left|\partial_{y}I_{t}\right|}$$
where $\partial_{x}$ and $\partial_{y}$ are symbols for partial differentiation and $d_{t}^{*}=d_{t}/\bar{d_{t}}$ is the mean-normalized inverse depth from [40] to discourage shrinking of the estimated depth. The auto-masking of stationary points [11] in Equation (5) is used to overcome the issues of pixels that remain with the same appearance between neighboring frames. We therefore define the binary mask μ, i.e.,
(8)$$\mu =[\underset{t^{\prime }}{\min\;}pe\left(I_{t},I_{t^{\prime }\to t}\right)<\underset{t^{\prime }}{\min\;}pe\left(I_{t},I_{t^{\prime }}\right)]$$
where [] denotes the Iverson bracket. Furthermore, the mask can also filter out pixels that are infinitely far from the camera. The final training loss is computed as the weighted sum of the per-pixel photometric loss and smoothness regularization term:(9)$$L=L_{p}+\lambda L_{smooth}$$

#### 3.3.2. Hard Attention for Multi-Scale Estimation

Many existing efforts have demonstrated that using multi-scale depth prediction can effectively prevent training from falling into local optima. The previous models [14,21] here are a simple combination of prediction results at different scales in the depth decoder, as shown in Figure 4. Before image reconstruction at each scale, the resolution of the RGB image needs to be reduced to the same as the output depth map of each decoder layer. However, this method always creates ‘holes’ in the low-texture regions of the intermediate depth maps that have a low resolution due to the uncertainty of photometric error. In order to overcome the issue of holes, the Monodepth2 model upsamples the lower-resolution depth maps to the original RGB image resolution and finally computes the photometric loss at the input resolution. Inspired by techniques in stereo reconstruction [41], the process here is effective because the depth values in the low-resolution depth map can be responsible for the corresponding ‘patch’ of pixels in the high-resolution RGB image. After computing the per-pixel photometric loss at each scale, Monodepth2 combines the multi-scale photometric losses as:(10)$$L_{mp}=\frac{1}{\left|I\right|}\underset{i\in I}{\sum }L_{p}{}^{\left(i\right)}$$
where $I=\left\{0,\;1,\;2,\;3\right\}$ is the set of different scales shown in Figure 4 and $L_{p}$ can be computed by Equation (5).

According to Equation (10), we can easily conclude that the new depth maps obtained by different upsampling factors have exactly the same importance degree as the weights of photometric losses at different scales that are equal to 1 when calculating the loss $L_{mp}$. In Figure 5 we show a simple example of computing the loss $L_{mp}$ for two scales: $scale_{0}$ and $scale_{2}$ in our depth decoder. We can observe that $scale_{0}$ has no upsampling since it has the same size as the input image, and $scale_{2}$ has an upsampling factor of 4, which means 3/4 of the pixels in $scale_{2}$ are artifacts. We can finally calculate the multi-scale photometric loss $L_{mp}$ after averaging over each pixel and scale. However, it does not seem very reasonable that depth maps with various levels of noise or different amounts of depth artifacts have the same responsibility for the loss $L_{mp}$. Therefore, as shown in Figure 4, we designed a hard attention strategy in our SHdepth model to overcome this drawback. Specifically, we proposed a new weight assignment based on the upsampling factors for multi-scale depth estimation. Our final photometric loss $L_{fp}$ is computed as the weighted sum of losses at different scales:(11)$$L_{fp}=\frac{1}{\left|I\right|}\underset{i\in I}{\sum }(2^{-i}\times L_{p}{}^{\left(i\right)})$$

Furthermore, the smoothness regularization term also adopts the hard attention strategy in our SHdepth. Therefore, our final training loss is computed as:(12)$$L=\frac{1}{\left|I\right|}\underset{i\in I}{\sum }[2^{-i}\times (\lambda_{1}L_{p}{}^{\left(i\right)}+\lambda_{2}L_{smooth}^{\left(i\right)})]$$

## 4. Experiments

In this section, we validate that (1) our soft attention module can significantly improve accuracy, especially when predicting sharper edges or thinner objects; (2) both spatial attention and channel attention in our soft attention module can improve the depth prediction results; and (3) the hard attention strategy implemented by designing a new loss function can also improve the results. We evaluate our SHdepth, mainly on the KITTI dataset [2], to allow the fair comparison with most existing self-supervised methods.

### 4.1. Implementation Details

We implemented our models by using the PyTorch library, and trained them mainly on a single Nvidia 2080Ti GPU with a batch size of 12. We trained the model on a single Nvidia 3090 GPU only when we replaced the backbone ResNet-18 by ResNet-50 due to the need for more memory. The depth and pose networks were jointly trained for 20 epochs using Adam [42] with $\beta_{1}=0.9$, $\beta_{2}=0.999$, and with an initial learning rate of $10^{-4},$ which was then decayed to $10^{-5}$ for the last 5 epochs. We also used weights pretrained on ImageNet [43] for our ResNet encoder as this has been shown to effectively improve accuracy in previous work [11]. The input images include three consecutive frames and were set to a fixed resolution of 640×192 during training. The training time of our SHdepth model was about 17 h and 34 min and the inference time of a single image was about 0.142 s. The final training loss in Equation (12) was computed with the photometric term $\lambda_{1}$ set to 32/15 and the smoothness term $\lambda_{2}$ set to 0.001. It is worth mentioning that $\lambda_{1}$ was set to that value because we tend to have the same ratio of multi-scale photometric losses to smoothness losses in our model as in Monodepth2.

Our model consists of two parts: the depth network and the pose network. In Table 1 and Table 2 we describe the details of our depth network and pose decoder network, including the parameters, the input and the activation function of each layer. Furthermore, the details of our soft attention module can be found in Section 3.2. We implemented SHdepth with ResNet as a shared encoder, which was integrated in both depth network and pose network. For all experiments, except where stated, we used ResNet-18 as our shared encoder. We fed the feature maps of three consecutive frames generated by ResNet to the decoder of our pose network. The outputs of our pose network are two relative poses including translations and rotations. Unlike the training process, we only used the maximum output scale of our depth decoder for evaluation.

### 4.2. KITTI Results

As with most previous papers, we used the KITTI benchmark for training and evaluation. We also adopted the data split of Eigen et al. [44] and removed static frames with Zhou et al.’s [7] pre-processing. Eventually, we used 39,810 monocular triplets for training, 4424 for validation, and 697 for evaluation. For all the images in KITTI, we used the same intrinsics. Specifically, we set the principal point to the image center and the focal length to the average of all the focal lengths in the dataset. Since self-supervised training based on monocular video has no scale information, we have reported the results using the per-image median ground-truth scaling [7].

We evaluated our depth prediction by using the metrics described in [44]. As shown in Table 3, we compared our method with other representative works. All the self-supervised approaches can be divided into two categories according to whether the input at the time of testing is a single frame or multiple frames. In general, the methods where the input is a single frame at test time are more widely used, although there is a performance gap between them and multi-frame methods. The results in Table 3 show that our method outperforms all existing approaches and further closes the gap between single-frame methods and multi-frame methods. We observe that our SHdepth with ResNet-50 has the best performance. Furthermore, our SHdepth can beat all the methods except HR-Depth when we use ResNet-18 as the backbone. Despite the fact that Packnet-SFM has better values for evaluation measures Sq Rel and RMSE, the number of its parameters is 128.29 M, which is far more than our SHdepth (R18) with 15.66 M parameters.

To show the effect of our model more visually, we compared some qualitative experimental samples with other methods, which can be seen in Figure 6. In general, there is a common issue with all self-supervised approaches when the scene contains transparent surfaces such as the car windows. The depth predicted will be the car interior rather than the surface of the window. However, qualitative results show that our method has sharper edges than Monodepth2, such as road signs and tree trunks. It also has slightly better results than PackNet-SfM, while having far fewer parameters. This probably benefits from our soft attention module, which captures long-range dependencies and explores the importance of different channels.

### 4.3. KITTI Ablation

In this section, we want to verify the impact of the components of our contribution on the overall performance, including the soft attention module and the hard attention strategy. In addition, we also conducted separate experiments for spatial attention and channel attention in the soft attention module. It merits mentioning that the channel attention here integrates the residual connections in the module. Therefore, the components of our model in the ablation experiment consist of three parts: spatial attention, channel attention, and hard attention.

In Table 4 we can see a full ablation study on the KITTI dataset, turning on and off the components of our model in various orders. We can conclude from the first three rows that the hard attention strategy can significantly improve the performance. The first row in Table 4 represents the previous models in Figure 4, which do not use the full-resolution multi-scale sampling method in Monodepth2. The second row is our baseline Monodepth2. Then, we can observe that nearly all the metrics have improved by using our hard attention strategy in the third row. From the fourth to the seventh row, it can be concluded that spatial attention and channel attention can achieve a steady improvement in performance regardless of whether a hard attention strategy is used or not, respectively. When the eighth row is compared with the second row, it can be seen that our soft attention module is effective in improving almost every evaluation measure. Finally, the last row shows that all our components of SHdepth lead to a remarkable improvement compared to Monodepth2 when combined together. We also compared some qualitative samples of ablation experiments, as shown in Figure 7. The baseline model represents the Monodepth2 method. Both our baseline and SHdepth here use ResNet-18 as the backbone of the encoder. The qualitative results show that our SHdepth is able to estimate the edges and details of some objects more accurately.

### 4.4. Make3D Results

To further validate our model, we performed additional experiments on the Make3D dataset [23]. Table 5 shows the quantitative results for the Make3D dataset using our SHdepth trained on the KITTI dataset. We can observe that our method has a more superior result than other self-supervised methods. To be more specific, we used our model to test on all 134 test images in Make3D without any additional training. We followed the same pre-processing method and testing protocol as Monodepth2. Firstly, we cropped all the images in the test dataset to a 2:1 ratio and then we also used the median scaling for our model at test time. Finally, we used the evaluation criteria outlined in [14] to compare with previous methods. Furthermore, we did not compare with methods [20] that use far more complex backbones than our ResNet-18 such as ResNet-101 for the sake of a fair comparison.

Qualitative results can be seen in Figure 8. We mainly compared some qualitative experimental samples with Monodepth2. Since the depth values of ground truth were sparse, we used interpolation to obtain dense depth maps. In fact, the depth maps generated by ground truth here were not as good as in KITTI because of the poor quality of the ground truth in Make3D. Qualitative results show that our SHdepth can successfully predict some detailed parts of the scene such as leaves, tree trunks, and small objects in the distance, which fail when predicted with the Monodepth2 model. This is likely because of the enhanced feature extraction with our soft attention module.

## 5. Conclusions

In this paper, we have presented a new method, SHdepth, to improve the accuracy of self-supervised monocular depth estimation. By incorporating a soft attention module, we further enhanced the extraction of the corresponding features by the depth encoder network. Additionally, we introduced a hard attention strategy into the loss function calculation after analyzing the drawbacks of the full-resolution multi-scale sampling method in Monodepth2. The two contributions mentioned above, the soft attention module and the hard attention strategy, act on different stages of the decoder, one for the input and the other for the decoding process. They have successfully allowed us to estimate a more accurate depth when combined together. The experiments have demonstrated that our approach can outperform other existing self-supervised methods.

In future work, we plan to investigate how to improve the depth estimation for moving objects through an attention mechanism. Furthermore, we will focus on adaptive weight assignment for multi-scale depth maps, which may be superior to our hard attention strategy.

## Figures and Tables

**Figure 1 sensors-21-06956-f001:**
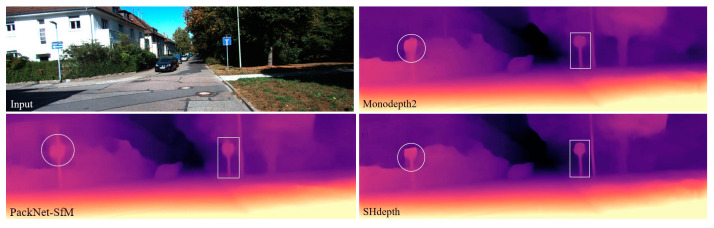
Depth prediction from a single image. We compared our self-supervised method, SHdepth, with Monodepth2 [11] and PackNet-SfM [17]. Our model produces a higher quality and sharper depth map.

**Figure 2 sensors-21-06956-f002:**
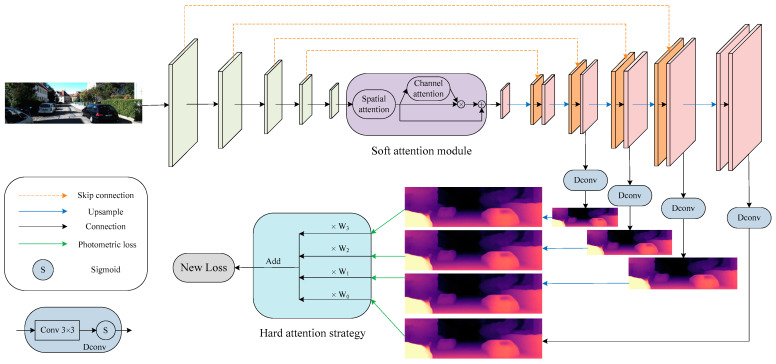
Overall architecture. Our depth network is based on a standard U-Net composed of a ResNet encoder and a depth decoder. Our soft attention module is integrated between the encoder and the decoder. We also use skip connection between encoding layers and associated decoding layers. The depth is decoded by our Dconv block, which consists of a 3×3 convolution and a sigmoid function. Furthermore, our hard attention strategy is employed after computing multi-scale photometric losses during training.

**Figure 3 sensors-21-06956-f003:**
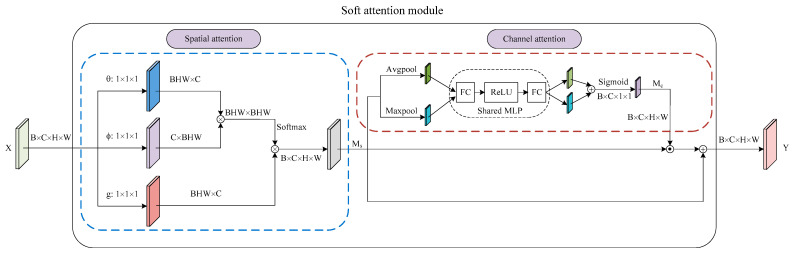
Illustration of soft attention module. We show the arrangement of the spatial and channel attention and more details of these two submodules.

**Figure 4 sensors-21-06956-f004:**
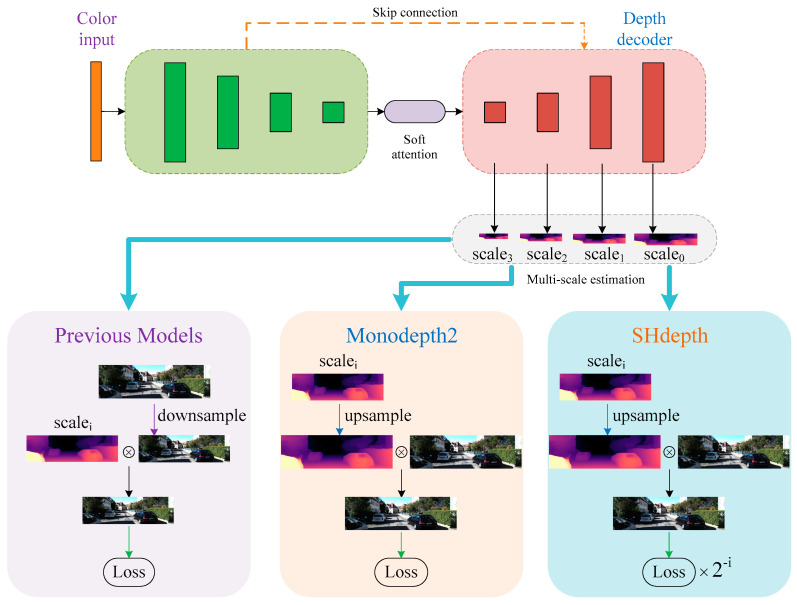
Hard attention strategy in SHdepth. We compared the details of our hard attention strategy with Monodepth2 and other previous methods. All of them utilize information from multi-scale depth maps during training.

**Figure 5 sensors-21-06956-f005:**
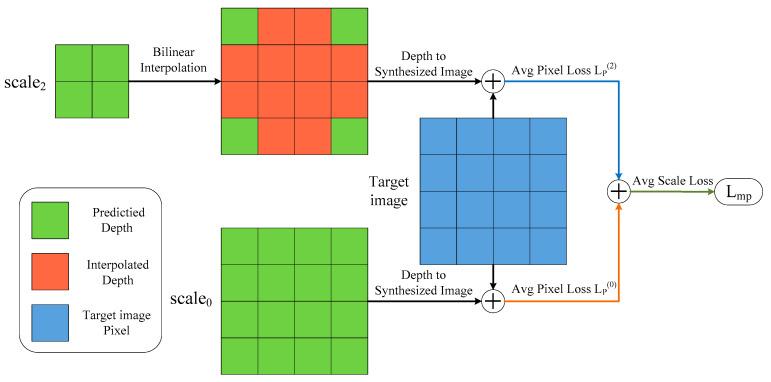
Analysis of the problem with calculating $L_{mp}$ in Monodepth2. Here we take only two of the four depth maps with different scales for our analysis and we adopt bilinear interpolation to implement the upsampling of depth maps.

**Figure 6 sensors-21-06956-f006:**
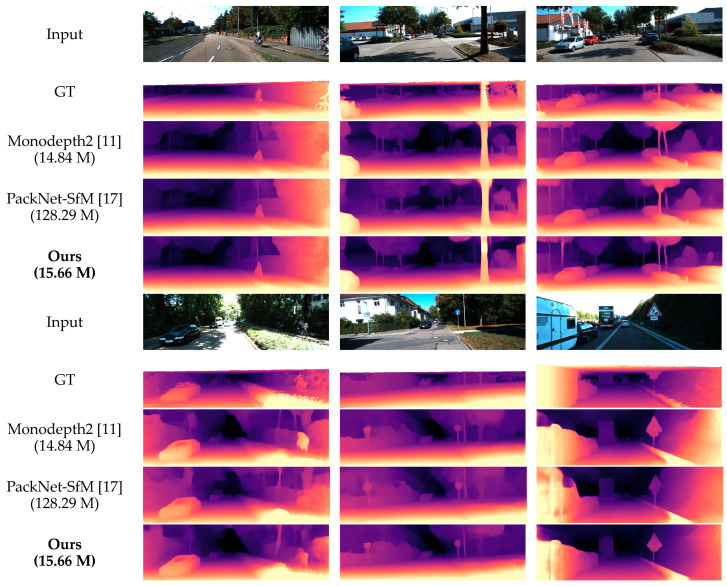
Qualitative results on the KITTI Eigen test split [44]. Ground truth depth maps (row 2 and row 7) are interpolated from sparse point clouds for visualization purposes. Here we use ResNet-18 as the backbone of our SHdepth encoder.

**Figure 7 sensors-21-06956-f007:**
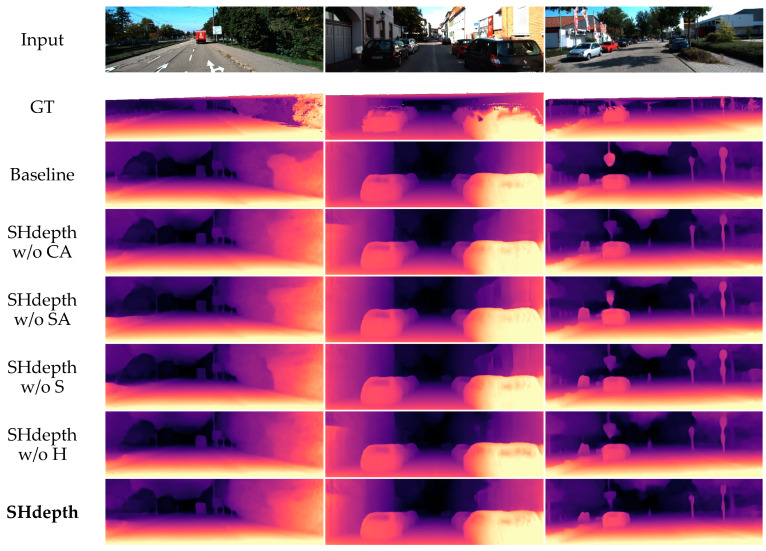
Qualitative ablation study on the KITTI eigentest split. CA (row 4) is channel attention and SA (row 5) is spatial attention. S (row 6) is the soft attention module and H (row 7) is the hard attention strategy in our SHdepth.

**Figure 8 sensors-21-06956-f008:**
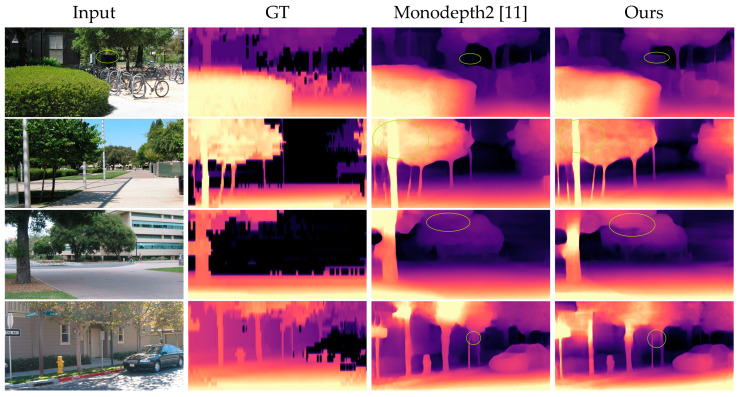
Qualitative samples on the Make3D datasets. We can conclude from the green-ellipse-marked areas that our method significantly outperforms Monodepth2 in these samples.

**Table 1 sensors-21-06956-t001:** Depth network. Here k is the kernel size and s is the stride. Chns denotes the number of output channels and res is the resolution scale, which represents the downscaling factor relative to the resolution of the input image. The symbol ↑ is a 2× nearest-neighbor upsampling of the layer. The ‘none’ in the table means that there is no activation function in this layer.

Layer	k	s	chns	res	Input	Activation
conv1	7	2	64	2	image	ReLU
maxpool	3	2	64	4	conv1	none
res1	3	1	64	4	maxpool	ReLU
res2	3	2	128	8	res1	ReLU
res3	3	2	256	16	res2	ReLU
res4	3	2	512	32	res3	ReLU
Spatial attention	N/A	N/A	512	32	res4	N/A
Channel attention	N/A	N/A	512	32	Spatial attention	N/A
upconv4	3	1	256	32	Channel attention	ELU
skipconv4	3	1	256	16	upconv4↑, res3	skipconv4
upconv3	3	1	128	16	skipconv4	ELU
skipconv3	3	1	128	8	upconv3↑, res2	ELU
scale3	3	1	1	1	skipconv3	Sigmoid
upconv2	3	1	64	8	skipconv3	ELU
skipconv2	3	1	64	4	upconv2↑, res1	ELU
scale2	3	1	1	1	skipconv2	Sigmoid
upconv1	3	1	32	4	skipconv2	ELU
skipconv1	3	1	32	2	upconv1↑, conv1	ELU
scale1	3	1	1	1	skipconv1	Sigmoid
upconv0	3	1	16	2	skipconv1	ELU
newconv0	3	1	16	1	upconv0↑	ELU
scale0	3	1	1	1	newconv0	Sigmoid

**Table 2 sensors-21-06956-t002:** Pose decoder network. The ‘squeeze*3’ means that the input layer of pconv0 is the result of concatenating three squeeze layers, which correspond to three consecutive images as the initial input of our encoder.

Layer	k	s	chns	res	Input	Activation
squeeze	1	1	256	32	res4	ReLU
pconv0	3	1	256	32	squeeze*3	ReLU
pconv1	3	1	256	32	pconv0	ReLU
pconv2	1	1	12	32	pconv1	none

**Table 3 sensors-21-06956-t003:** Quantitative results. Comparison of our SHdepth to other existing methods on KITTI. Best results in each category are in bold and second best are underlined. The R18 in the method column means that the encoder is ResNet-18 and R50 is ResNet-50. For error evaluating metrics Abs Rel, Sq Rel, RMSE, and RMSE log, lower is better. For accuracy evaluating metrics δ < 1.25, δ < 1.25^2^ and δ < 1.25^3^, higher is better.

Method	Test Frames	Abs Rel	Sq Rel	RMSE	RMSE Log	δ < 1.25	δ < 1.25^2^	δ < 1.25^3^
GLNet [45]	3 (−1, 0, +1)	0.099	0.796	4.743	0.186	0.884	0.955	0.979
Luo et al. [46]	N	0.130	2.086	4.876	0.205	0.878	0.946	0.970
Struct2depth (M+R) [27]	3 (−1, 0, +1)	0.109	0.825	4.750	0.187	0.874	0.958	**0.983**
Patil et al. [32]	N	0.111	0.821	4.650	0.187	0.883	0.961	0.982
Wang et al. [47]	2 (−1, 0)	0.106	0.799	4.662	0.187	0.889	0.961	0.982
CoMoDA [48]	N	0.103	0.862	4.594	0.183	0.899	0.961	0.981
McCraith et al. [49]	2 (0, +1)	**0.089**	0.747	4.275	0.173	0.912	0.964	0.982
ManyDepth [33]	2 (−1, 0)	0.090	**0.713**	**4.261**	**0.170**	**0.914**	**0.966**	**0.983**
Ranjan et al. [50]	1	0.148	1.149	5.464	0.226	0.815	0.935	0.973
EPC++ [13]	1	0.141	1.029	5.350	0.216	0.816	0.941	0.976
Struct2depth (M) [27]	1	0.141	1.026	5.291	0.215	0.816	0.945	0.979
Videos in the wild [28]	1	0.128	0.959	5.230	0.212	0.845	0.947	0.976
Li et al. [51]	1	0.130	0.950	5.138	0.209	0.843	0.948	0.978
Guizilini et al. (R18) [15]	1	0.117	0.854	4.714	0.191	0.873	**0.963**	0.981
Monodepth2 [11]	1	0.115	0.903	4.863	0.193	0.877	0.959	0.981
ASANet (R18) [19]	1	0.112	0.866	4.693	0.189	0.881	0.961	0.981
Johnston et al. (R18) [20]	1	0.111	0.941	4.817	0.189	**0.885**	0.961	0.981
PackNet-SfM [17]	1	0.111	**0.785**	**4.601**	0.189	0.878	0.960	0.982
HR-Depth [31]	1	**0.109**	0.792	4.632	**0.185**	0.884	0.962	**0.983**
**SHdepth (R18)**	1	0.111	0.828	4.692	0.187	0.881	0.961	0.982
**SHdepth (R50)**	1	**0.108**	0.812	4.634	**0.185**	**0.887**	0.962	0.982

**Table 4 sensors-21-06956-t004:** Ablation study. Results for different variants of our model (SHdepth) with monocular training on KITTI. Best results in each category are in bold. In the first row, ‘full-res m.s.’ is the full-resolution multi-scale sampling method in Monodepth2, as shown in Figure 4. Here the baseline model is Monodepth2. In the last row, our complete SHdepth is expressed as ‘Baseline + SA + CA + H’, where SA is spatial attention, CA is channel attention, and H represents the hard attention strategy.

Ablation	Abs Rel	Sq Rel	RMSE	RMSE Log	δ < 1.25	δ < 1.25^2^	δ < 1.25^3^
Baseline w/o full-res m.s.	0.117	0.866	4.864	0.196	0.871	0.957	0.981
Baseline (MD2)	0.115	0.903	4.863	0.193	0.877	0.959	0.981
Baseline + H	0.113	0.865	4.809	0.191	0.878	0.960	0.981
Baseline + SA	0.112	0.848	4.736	0.188	0.879	**0.961**	**0.982**
Baseline + CA	0.114	0.902	4.855	0.192	0.878	0.959	0.981
Baseline + H + CA	0.113	0.854	4.771	0.190	0.880	0.960	**0.982**
Baseline + H + SA	0.112	**0.826**	4.700	0.189	0.878	**0.961**	**0.982**
Baseline + SA + CA	**0.111**	0.833	4.704	0.189	**0.881**	0.960	0.981
**Baseline + SA + CA + H (Ours)**	**0.111**	0.828	**4.692**	**0.187**	**0.881**	**0.961**	**0.982**

**Table 5 sensors-21-06956-t005:** Make3D results. Best results in each category are in bold. D—depth supervision, S—self-supervised stereo supervision, M—self-supervised mono supervision, MS—self-supervised mono and stereo supervision. All M methods use the median scaling and the MS method predicts the depth by the unmodified network.

Method	Type	Abs Rel	Sq Rel	RMSE	RMSE Log
Karsch [52]	D	0.428	5.079	8.389	0.149
Liu [53]	D	0.475	6.562	10.05	0.165
Laina [5]	D	0.204	1.840	5.683	0.084
Monodepth [14]	S	0.544	10.940	11.760	0.193
Monodepth2 [11]	MS	0.374	3.792	8.238	0.201
Zhou [7]	M	0.383	5.321	10.470	0.478
DDVO [8]	M	0.387	4.720	8.090	0.204
Monodepth2 [11]	M	0.322	3.589	7.417	**0.163**
**SHdepth (Ours)**	M	**0.320**	**3.031**	**7.280**	**0.163**

## Data Availability

Publicly available datasets were analyzed in this study. The KITTI data set can be found here: http://www.cvlibs.net/datasets/kitti/raw_data.php (accessed on 9 September 2020); the Make3D data set can be found here: http://make3d.cs.cornell.edu/data.html (accessed on 10 August 2021).

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
