# Peer review of "Joint Soft–Hard Attention for Self-Supervised Monocular Depth Estimation"

_sensors, 2021, doi:10.3390/s21216956_

Round 1

Reviewer 1 Report

I've found this job rubust and of interest. I just have some minor concerns like double check the formality og eq. in Sec. 3.2.1. Spatial Attention.

I would also clarify a bit more the interaction between the soft and hard attention strategies, which sometimes goes a little out of focus.

Reviewer 2 Report

The paper presents a method for improving upon the current state-of-the-art on self-supervised monocular depth estimation. The authors employ a soft attention module between the network encoder and decoder and a hard attention strategy for multi-scale fusion.

Reported results show an improvement for monocular depth estimation. However, the improvement is rather small, and Table 4 shows that the hard attention strategy is only marginally improving upon the Baseline + SA + CA. In fact, the improvement might be small enough to be attributed to different random initializations.

However, qualitative results do show an improvement in some scenarios, but it is hard to judge based only on that.

Reviewer 3 Report

Authors propose a self-supervised monocular depth estimation approach based in soft and hard attention, intergrating attention in the model architecture. The methods have been properly described and implemented and diagrams have also been provided. 

The proposed approached was benchmarked against other SOTA approaches showing very promising resulting in a number of benchmarks datasets. 

Comments:

1. line 29: more references are needed.

2. line 47: what is meant by "A few special pixels"?

3. Provide a more details discussion on the proposed method performance with respect to other methods appearing in Table 3. The results appear to be mixed, so what is the advantage of this approach compared to the others?

4. Please provide some information on training and inference run times and also number of parameters for the end-to-end architecture.

Round 2

Reviewer 2 Report

Ok